# Extracellular Nucleotides Affect the Proangiogenic Behavior of Fibroblasts, Keratinocytes, and Endothelial Cells

**DOI:** 10.3390/ijms23010238

**Published:** 2021-12-27

**Authors:** Edyta Węgłowska, Maria Koziołkiewicz, Daria Kamińska, Bartłomiej Grobelski, Dariusz Pawełczak, Marek Kołodziejczyk, Stanisław Bielecki, Edyta Gendaszewska-Darmach

**Affiliations:** 1Institute of Molecular and Industrial Biotechnology, Faculty of Biotechnology and Food Sciences, Lodz University of Technology, Stefanowskiego 2/22, 90-537 Lodz, Poland; edytalaska4@gmail.com (E.W.); maria.koziolkiewicz@p.lodz.pl (M.K.); daria.kaminska@dokt.p.lodz.pl (D.K.); marek.kolodziejczyk@p.lodz.pl (M.K.); stanislaw.bielecki@p.lodz.pl (S.B.); 2Department of Experimental Surgery, Medical University of Lodz, 93-513 Lodz, Poland; bartlomiej.grobelski@umed.lodz.pl (B.G.); darekpawelczak@o2.pl (D.P.)

**Keywords:** extracellular nucleotides, angiogenesis, purinergic receptors, phosphorothioate analogs

## Abstract

Chronic wound healing is currently a severe problem due to its incidence and associated complications. Intensive research is underway on substances that retain their biological activity in the wound microenvironment and stimulate the formation of new blood vessels critical for tissue regeneration. This group includes synthetic compounds with proangiogenic activity. Previously, we identified phosphorothioate analogs of nucleoside 5′-O-monophosphates as multifunctional ligands of *P2Y6* and *P2Y14* receptors. The effects of a series of unmodified and phosphorothioate nucleotide analogs on the secretion of VEGF from keratinocytes and fibroblasts, as well as their influence on the viability and proliferation of keratinocytes, fibroblasts, and endothelial cells were analyzed. In addition, the expression profiles of genes encoding nucleotide receptors in tested cell models were also investigated. In this study, we defined thymidine 5′-O-monophosphorothioate (TMPS) as a positive regulator of angiogenesis. Preliminary analyses confirmed the proangiogenic potency of TMPS in vivo.

## 1. Introduction

Transmembrane G protein-coupled receptors (GPCRs) constitute a diverse and broad superfamily of proteins. Within GPCRs, significant academic and industry effort has been put into developing agonist and antagonist ligands for the *P2Y* receptors that respond to extracellular nucleotides. The *P2Y* receptors with eight subtypes (*P2Y1*, *P2Y2*, *P2Y4*, *P2Y6*, *P2Y11*, *P2Y12*, *P2Y13*, and *P2Y14*) recognized so far belong to the δ-branch of class A GPCRs [1]. ATP is an agonist for *P2Y2* and *P2Y11* receptors; ADP activates *P2Y1*, *P2Y12*, and *P2Y13*. UTP recognizes *P2Y2* and *P2Y4* while UDP is a ligand for the *P2Y6* receptor, and a nucleotide sugar conjugate (UDP-glucose) is an agonist for the *P2Y14*. Based on their sequence similarity, the *P2Y* receptors have been grouped into two subfamilies, namely *P2Y1*, *P2Y2*, *P2Y4*, *P2Y6*, and *P2Y11* coupled directly to Gq protein and *P2Y12*
*P2Y13*, and *P2Y14* coupled to Gi/o protein. *P2Y11* receptors’ activation also stimulates adenylate cyclase to generate cAMP [2].

An increasing number of experimental data supports the idea that a range of native *P2Y* ligands is major regulators of vascular cell functions. Especially, *P2Y1*, *P2Y2*, and *P2Y12* activation were shown to be implicated in the process of angiogenesis, an essential aspect of many non-healing wound [3,4].

In intact tissues, the microvasculature remains in a state of homeostasis to supply oxygen and nutrients and remove carbon dioxide and waste products. Upon injury, the microvasculature is disrupted, leading to fluid accumulation, inflammation, and the development of hypoxia [5]. Additionally, the healing process may vary depending on the type of wound and additional unfavorable conditions. Treatment of chronically healing wounds is a serious medical problem due to the prevalence and frequent complications. It primarily affects elderly patients, seriously limiting their quality of life. Chronic wounds, including ulcers and bedsores, are also a serious complication of many diseases associated with the local reduction of blood flow in venous and arterial vessels and microcirculation. Arterial hypertension, obesity, infections, and diabetes may also be contributing factors. Hyperglycemia reduces the proliferation rate of keratinocytes, fibroblasts, and endothelial cells, which are found in the skin and are therefore directly involved in the healing process. Moreover, hyperglycemic conditions favor the increase in reactive oxygen species (ROS) [6]. Reduced angiogenesis in chronic venous ulcers was also associated with a decreased expression of various angiogenic growth factors, especially vascular endothelial growth factor (VEGF-A) [6], increased proteolysis of VEGF-A [7], as well as increased levels of soluble VEGF receptor, which may serve to neutralize VEGF-A activity [8].

Great progress has allowed for the synthesis of many healing-accelerating molecules intending to induce therapeutic angiogenesis. In particular, growth factors were a promising therapeutic strategy since in vivo these proteins play a critical role in the wound healing process. Unfortunately, many growth factor-based therapies have not shown clear benefits in clinical trials. The limited clinical efficacy of protein growth factors is associated with inactivation by the local action of proteases [9] or the development of resistance to angiogenic growth factors during the development of vascular disease and the aging process [10]. Recent advances in our understanding of chronic wound biology have led to the development of small-molecular therapies that promote angiogenic activity and modulate the repair process with exciting potential for clinical application [11]. Many literature data show that nucleotides secreted from cells into the extracellular space in response to damage take part in all stages of wound healing by activating the appropriate nucleotide receptors [12]. However, the therapeutic potential of unmodified nucleotides is limited due to the rapid degradation by numerous extracellular enzymes, which significantly reduces their duration of action and thus also their effectiveness. These enzymes include members of ecto-NTPDases (ecto-nucleoside triphosphate diphosphohydrolases), ecto-NPPs (ecto-nucleotide pyrophosphatase/phosphodiesterase), ecto-alkaline phosphatases, and ecto-5′-nucleotidases [13]. To overcome these limitations, attempts have been made to increase the stability of nucleotides, e.g., by synthesizing phosphorothioate analogs of nucleotides. The studies on the stability of phosphorothioate analogs of nucleotides have shown that replacing an oxygen atom with sulfur in a phosphate group conferred significant resistance to enzymatic degradation by extracellular enzymes [14].

We have shown previously that nucleoside 5′-O-phosphorothioate analogs, namely UTPγS, UTPαS, and ATPγS, accelerated VEGF release and migration of human HaCaT keratinocytes. The potency of phosphorothioate analogs of ATP and UTP correlated with the highest *P2Y2* receptor expression by the HaCaT cell line [15]. In the present study, we tested the ability of unmodified nucleotides (ADP, ATP, UDP, and UTP) and their phosphorothioate analogs (ADPβS, UDPβS, ATPγS, and UTPγS) to stimulate secretion of VEGF, a key factor in angiogenesis, from keratinocytes and fibroblasts, as well as their influence on the process of survival and proliferation of keratinocytes, fibroblasts and endothelial cells. The selection of these cell types was determined by their significant participation in the angiogenesis and wound healing processes [16]. The presented hereby research was extended to 5′-monophosphorothioate analogs (AMPS, CMPS, TMPS, and UMPS) since we have shown that nucleoside 5′-O-monophosphorothioates can also act as *P2Y* ligands [17,18]. The expression pattern of P2 receptors quantified by quantitative RT-PCR analysis was also estimated.

## 2. Materials and Methods

### 2.1. Tested Compounds

ADP, ATP, UDP, and UTP were purchased from Sigma–Aldrich (Merck KGaA, Darmstadt, Germany). Phosphorothioate analogs of nucleotides (TMPS, AMPS, CMPS, UMPS, ADPβS, UDPβS, ATPαS, ATPγS, UTPαS, UTPγS) were obtained from BioLog (Bremen, Germany).

### 2.2. Cell Cultures

All cells were cultured under standard conditions at 37 °C in a humidified atmosphere of 95% air and 5% CO_2_. Most of the culture reagents were obtained from Life Technologies (Carlsbad, CA, USA). Otherwise, the source is given in parentheses. Moreover, the culture media were supplemented with antibiotics, namely 100 U/mL of penicillin (Polfa Tarchomin, Warsaw, Poland) and 100 µg ml of streptomycin (Polfa Tarchomin, Warsaw, Poland). 

#### 2.2.1. HaCaT Cell Line

The immortal human keratinocyte HaCaT cell line was purchased from Leibniz Institute DSMZ-German Collection of Microorganisms and Cell Cultures (Braunschweig, Germany). Cells were cultured in Dulbecco’s modified Eagle’s medium (DMEM) supplemented with 10% fetal calf serum (FCS) containing 100 IU/mL of penicillin and 100 µg/mL of streptomycin.

#### 2.2.2. HUVECs

Human umbilical vein endothelial cells (HUVECs) were purchased from Cascade Biologics, Inc. (Portland, OR, USA). Cells were grown in culture vessels coated with a 1% gelatin (Sigma–Aldrich) solution dissolved in phosphate-buffered saline (PBS, Thermo Fisher Scientific, Waltham, MA, USA). Growth medium consisted of a mixture of RPMI 1640 medium with the addition of 20% fetal bovine serum (FBS) and LSGS (low serum growth supplement) in the amount of 1 mL per 50 mL of medium. The final concentrations of LSGS substances in the culture medium were as follows: 2% FBS, 1 μg/mL of hydrocortisone, 10 ng/mL of hEGF (human epidermal growth factor), 3 ng/mL of basic fibroblast growth factor bFGF (basic fibroblast growth factor), 10 μg/mL of heparin.

#### 2.2.3. Human Dermal Fibroblasts

Human dermal fibroblasts, adult (HDF) were acquired from the Life Technologies collection. Cells were grown in Medium 106 supplemented with LSGS in the amount of 1 mL per 50 mL of medium.

### 2.3. Expression of P2Y Genes with Reverse Transcription Quantitative Polymerase Chain Reaction

Extraction of the total RNA from HUVECs and HDFs was performed with the RNeasy^®^ Mini Kit (QIAGEN, Venlo, The Netherlands). Subsequently, RNA was purified with Amplification Grade DNase I (Sigma-Aldrich) and then reverse-transcribed to with RT^2^ First Strand Kit (SABioscences, Frederick, MD, USA). All primers were designed using the NCBI’s Primer-BLAST software (National Center for Biotechnology Information, Bethesda, MD, USA) based on *P2Y* sequences from the GenBank database (Table 1) and purchased from Genomed (Warsaw, Poland). Real-time RT-PCR was conducted using SYBR^®^ Green-based RT^2^ qPCR Master Mix (SABioscences, Frederick, MD, USA) with a Chromo 4 detection system (Hercules, Bio-Rad, CA, USA). cDNA representing 6 ng per sample of total RNA was subjected to 40 cycles of PCR amplification. Samples were first incubated at 95 °C for 15 s, then at 60 °C for 30 s, and finally at 72 °C for 30 s. The PCR products were electrophoretically separated in a 1.0% agarose gel in Tris/borate/EDTA containing 0.5 µg/mL ethidium bromide. Housekeeping *GAPDH* gene was were selected as endogenous controls to correct potential variation in RNA loading. Expression of nucleotide receptors was normalized to GAPDH and the *P2Y6* receptor was chosen as a reference calibrator for giving relative expression values. The amount of target gene expression level was calculated with Livak’s method as 2^−ΔΔCt^, where ΔΔC_t_ = [C*_t_*(target) − C*_t_*(GAPDH)]sample − [C*_t_*(target) − C*_t_*(GAPDH]calibrator.

### 2.4. Cellular Metabolic Activity

Cells were seeded into 96-well cell culture plates at a density of 1 × 10^4^ cells/well in 100 μL of complete culture medium. On the following day, cells were washed with PBS and 100 μL of fresh serum-free medium was added. Subsequently the fasting medium was supplemented with a respective nucleotide added to a final concentration of 100 µM and 48 h later 10 μL of Presto Blue^®^ cell viability reagent (Life Technologies, Carlsbad, CA, USA) was applied. Following 80 min incubation time, cell viability was determined by measuring the fluorescent signal (Ex/Em = 530/590 nm) on a Synergy 2 microplate reader (BioTek, Winooski, VT, USA). The obtained fluorescence magnitudes were used to calculate cell viability expressed as a percent of the viability of the untreated control cells.

### 2.5. Proliferative Activity (DNA Quantification Assay)

DNA content was measured using a CyQUANT Direct Cell Proliferation Assay (Life Technologies). CyQUANT detection reagent dye was added to the wells in a volume of 50 μL, incubated for 60 min and the fluorescence signal (Ex/Em = 485/528 nm) detected using a Synergy 2 microplate reader. The obtained fluorescence magnitudes were used to calculate cell proliferation expressed as a percent of the proliferation of the untreated control cells.

### 2.6. Quantification of VEGF-A Secretion

Human VEGF protein was measured in culture media using the VEGF Human ELISA Kit (Life Technologies). Briefly, cells (3 × 10^4^ cells per well) were seeded into 96-well plate in complete medium. After 24 h of incubation cells were starved for another 24 h in the culture medium without serum and subsequently treated with tested nucleotides added to a final concentration of 100 µM for another 24 h. Finally, cell culture supernatants were collected and VEGF levels were determined according to the manufacturer’s protocol.

### 2.7. Determination of Angiogenesis In Vivo

All experimental procedures were approved by the Animal Bioethical Committee of Medical University of Lodz (no. 579/2011). Young adult Wistar rats (2–3 months; *n* = 6) were kept separately in cages before and after surgical procedures at constant room temperature in the animal house of Medical University of Lodz, Poland. Rats were fed a normal chow diet and kept under a controlled 12 h light/dark cycle. Bionanocellulose scaffolds were produced by *Komagataeibacter xylinus* in stationary conditions according to the procedure published previously [17]. BNC membranes were then cut into slices with dimensions of 1 × 1 × 0.3 cm, sterilized by autoclaving at 121 °C and then kept in PBS. Before implantation BNC membranes were soaked with sterile TMPS solution (0.7 mg/scaffold) or with PBS. The rats were weighed and anesthetized by administering ketamine (intramuscularly) at a concentration of 20 mg/kg *c.c*. The rat back hairs were shaved with the underlying skin cleaned and sterilized. To implant the scaffolds, two 1-cm incisions were cut on the dorsal section of each rat. Two BNC scaffold samples (control and with TMPS) were separately and implanted into each rat. The incisions were then sutured using MonosofTM 4/0 monofilament nylon sutures. All animals were then monitored by animal care services. After 30 days post implantation the rats were euthanized by giving an overdose of ketamine delivered as previously. The dorsal skin was carefully resected and sections containing cellulose scaffolds were photographed.

### 2.8. Statistical Analysis

Statistical analysis was performed using GraphPad Prism 6.0 software (GraphPad Software, Inc., La Jolla, CA, USA) and data are presented as the mean ± SD of at least three independent experiments. Statistical differences between two groups were analyzed by one-way ANOVA with Bonferroni post hoc test. Differences between groups were rated significant at a probability error *p* < 0.05 (*), *p* < 0.01 (**), *p* < 0.001 (***), and *p* < 0.0001 (****). *p* < 0.05 (*) was regarded as statistically significant.

## 3. Results

### 3.1. Expression of Genes Encoding P2Y Receptors in HUVECs and HDF Cells

So far, there are little data available in the literature on the quantitative analysis of the expression of nucleotide receptors in cells. Moreover, the relatively recently discovered *P2Y12*, *P2Y13*, and *P2Y14* receptors were mainly not considered in the studies. Our previous studies revealed that in human HaCaT keratinocytes, *P2Y2* showed the highest expression of all receptors studied. We observed about two times lower level of *P2Y6* and a small amount of *P2Y1* transcript. Expression of other *P2Y4* and *P2Y11*-*14* was also detected, but the distribution was much lower [15]. Hereby, RT-qPCR analyzes of *P2Y* genes were performed for human umbilical vein endothelial cells (HUVECs) and human primary fibroblasts isolated from adult skin (HDF) due to their crucial role in the angiogenesis process. The relative expression was compared with the *GAPDH* housekeeping gene and *P2Y6* was used as a calibrator (i.e., expression of each receptor was demonstrated as a ratio of *P2Y6*) to illustrate the expression of P2 receptors relative to each other.

We found that HUVECs expressed six *P2Y* subtypes (Figure 1A). The *P2Y6* gene shows the highest level of expression. The genes encoding *P2Y1* and *P2Y14* receptors were next in terms of intensity of expression, followed by *P2Y4*, *P2Y2,* and *P2Y11*. Expression of *P2Y12* and *P2Y13* genes was not detected. In HDF fibroblasts, the mRNA of all known *P2Y* receptors was detected (Figure 1B). As in the case of HUVECs, in HDFs the expression of *P2Y6* was seen at the highest level. However, these cells also expressed the *P2Y1* at a high level. A significantly lower expression of the *P2Y14* gene compared to *P2Y1* and *P2Y*6 and a significantly lower expression of the remaining *P2Y*s was also detected.

### 3.2. Cellular Metabolic Activity in the Presence of Extracellular Nucleotides

Our research used a rezasurin-based metabolic assay, the PrestoBlue^®^ test, to assess cell numbers after treatment with tested nucleotides. The resazurin-based assays use the mitochondrial activity to reduce the nonfluorescent blue resazurin to the fluorescent pink resorufin [18]. The results proved that unmodified nucleotides and their phosphorothioate analogs did not cause cytotoxicity against HUVEC endothelial cells, human dermal fibroblasts, and human HaCaT keratinocytes (Figure 2). In some cases, after 24 and 48 h of incubation, we even observed the stimulation of cell growth slightly exceeding 20% in the case of HUVECs and HDF. The highest stimulation of metabolic activity was observed in HaCaT cells and UTP, UTPαS, UTPγS, UDP, UDPβS, and ADPβS demonstrated the most increased activity.

### 3.3. Cellular Proliferation in the Presence of Extracellular Nucleotides

Our studies also employed the fluorometric CyQUANT^®^ assay (Life Technologies, Carlsbad, CA, USA) to determine DNA content reflecting proliferative activity, which is a critical feature regarding the survival of cells (Figure 3). It was shown in some studies that metabolic assays were used to quantify changes in cell growth that may not accurately reflect cellular proliferation rates [18]. The CyQUANT^®^ test contains a cyanine dye characterized by a low intrinsic fluorescence and significant fluorescence enhancements upon binding to nucleic acid [19].

Analysis of HUVECs proliferation showed that some of the nucleotides tested have a more significant impact on this process than the metabolic activity tested with the PrestoBlue^®^ assay. After 24 h, stimulation of proliferation exceeding 20% was caused by AMPS and ATP and its analog, ATPγS. After 48 h, ATP and ATPγS were the most active, inducing proliferation stimulation at 41% and 46%. A high level of proliferation was also demonstrated after incubating cells with ADP, ADPβS as well as some mononucleotide analogs, namely TMPS and UMPS. As in the case of the metabolic activity test, the proliferation of human HDF fibroblasts was only slightly stimulated by most of the nucleotides tested. The amount of synthesized DNA showed statistically significant changes in proliferation; however, the stimulation level was not high. No significant increase in the proliferation level of HDF fibroblasts was observed within 24 h. After 48 h of incubation of cells with compounds, results indicated that ATPαS and ATPγS increased the proliferation of these cells by around 30%. In addition, a slight effect of nucleotides on the expansion of human HaCaT keratinocytes was observed. After 24 h, only thymidine and uridine nucleotides and their analogs increased proliferation by approx. 20%. After 48 h, uridine-containing compounds showed the highest activity.

### 3.4. The Influence of Extracellular Nucleotides on the Secretion of the Proangiogenic Factor VEGF-A

To measure the quantity of VEGF in the conditioned medium from HaCaT cells, we employed an ELISA kit specific for the most frequent type of VEGF isoforms, namely VEGF-A. VEGF-A concentration was measured after 24 h of incubation with extracellular nucleotides. We found that under standard culture conditions, HDFs produced VEGF at the average level of 198.0 pg/mL/24 h per 3 × 10^4^ cells. Most of the nucleotides studied had a stimulating effect on the secretion of VEGF-A protein. Nucleoside monophosphorothioates appeared to be the most active in stimulating fibroblasts to secrete VEGF-A (Figure 4). The most potent TMPS and CMPS caused more than a twofold increase in the concentration of secreted VEGF-A, while their unmodified counterparts did not affect the secretion of this protein). A high increase was also observed after incubating the cells with AMPS, UMPS, UDPβS, and ATPγS. The remaining compounds did not cause statistically significant differences in the concentration of the analyzed proangiogenic factor (Figure 4A).

We previously showed that treatment with UTP, UTPαS, and UTPγS strongly induced VEGF production in HaCaT cells [15]. Therefore, in this work we also checked the potential of nucleoside monophosphates and monophosphorothioates to stimulate VEGF secretion in keratinocytes. Untreated HaCaT cells produced an average of 289.3 pg/mL/24 h per 3 × 10^4^ cells of VEGF-A. Incubation in the presence of specific nucleotides, especially their phosphorothioate analogs, significantly increased the production of VEGF-A. In the latter group, TMPS and AMPS had the greatest potency (Figure 4B) and caused an increase in the amount of VEGF-A released by approx. 100 pg/mL/24 h per 3 × 10^4^ cells, which is approx. 30% of the rise.

### 3.5. Thymidine 5′-O-Monophosphorothioate as a Proangiogenic Nucleotide

Considering the results of extracellular nucleotides’ ability to release VEGF, the most preferred proangiogenic compound turned out to be TMPS. Additionally, this compound showed no cytotoxicity, and in the case of HUVECs, it slightly stimulated cellular proliferation. Besides, in our previous studies, we identified TMPS as an inhibitor UDP-glucose (UDPG)-induced degranulation in a rat mast cell line (degranulation of RBL-2H3). On the other hand, other nucleoside 5′-O-monophosphorothioate tested (AMPS, CMPS, UMPS) did not inhibit degranulation of mast cells and even caused enhancement of *N*-acetyl-β-D-hexosaminidase (a granule enzyme that parallels histamine release). Thus, inhibitors selectively preventing mast cell activation might prospectively offer novel anti-inflammatory therapeutic approaches [20].

Therefore, we decided to get further insight into how TMPS affects the mechanism of angiogenesis. We focused on the expression of exemplary genes, which play a role during vascularization (*VEGFA*) and extracellular matrix remodeling (*MMP2*). MMP2 is an essential member of the matrix metalloproteinase family, which mediates extracellular matrix remodeling and is a prerequisite for angiogenesis [21]. We also included *MACF1* encoding microtubule actin cross-linking factor that plays an essential role in coordinating cell migration, cell proliferation, and maintenance of tissue integrity in the presence of F-actin and microtubules, critical for wound repair in skin epidermis [22]. We observed significant *MMP2* mRNA level upregulation when HUVECs were treated with TMPS. *MACF1* was also increased but to a less extend (Figure 5A). In turn, in human dermal fibroblasts, TMPS significantly increased mRNA levels of *VEGFA* (Figure 5B), supporting the results obtained for VEGF-A secretion in HDFs. MACF1 expression was also upregulated in both cell types studied.

Finally, we used TMPS to assess the possibility of promoting physiological host-mediated angiogenic response in vivo. Even though assays that monitor endothelial cell proliferation in in vitro culture benefit from being rapid, precise, and quantifiable, in vivo angiogenesis reflects a fully functional angiogenic process that works in tandem with various other processes. We employed a methodology based on implanting a polymer scaffold subcutaneously in the rat to monitor the angiogenic ingrowth. The scaffold material containing a test substance that should recruit host-derived endothelial cells is explanted at the end of the assay. Our studies used bionanocellulose (BNC) produced by *Komagataeibacter xylinus* as a scaffold because of its biocompatibility and high purity [23]. Two independent skin incisions about 1-cm-long were created on the back of Wistar rats and BNC with dimensions of 1 × 1 × 0.3 cm was implanted in each subcutaneous pouch (Figure 5C). After suturing the wound edges, each rat was placed in a clean cage and subjected to daily observation. No rats showed signs of pain that the BNC scaffold implantation could have caused, and none of the rats showed signs of inflammation or infection. Finally, 30 days after implantation, BNC preparations were removed for macroscopic observation. Healthy tissues were observed surrounding the BNC with the presence of blood vessels. In the macroscopic photos taken during the resection, it was seen that in the case of TMPS-soaked BNC (Figure 5E), blood vessels and capillaries could be observed more extensively within the surrounding dermal tissue. The network of blood vessels was dense, with some of the vessels growing into the preparation (small capillaries were dominant).

## 4. Discussion

Wound healing is a biological process that involves interactions between cells, extracellular matrix components, and growth factors. Under physiological conditions, four stages in the healing process can be distinguished, hemostasis, inflammatory phase, proliferative phase, and scar remodeling. The individual phases overlap in time and space [24]. Besides the proliferation of fibroblasts and keratinocytes, angiogenesis is critical for wound healing, particularly after chronic or ischemic damage. Newly created blood vessels provide nutrition and oxygen to cells at the wound site. VEGF is a crucial element in vascular development, as it promotes angiogenesis by increasing endothelial cell survival, proliferation, and migration [25]. VEGF has been widely investigated in the context of cutaneous wound healing and was shown to be substantially increased in healing wounds. Treatment with VEGF inhibitors or genetic deletion of *VEGF* in keratinocytes resulted in delayed wound closure, demonstrating the overall relevance of VEGF in wound healing [26]. As a result, developing new strategies for therapeutic angiogenesis remains a critical priority for treating a range of wound healing deficits.

Application of exogenous nucleotides was shown to promote wound healing via the activation of purinergic receptors [12]. Activation of *P2Y2* receptor by ATP or UTP was identified to induce the expression of the *VEGF* gene in fibroblasts [27]. In coronary artery endothelial cells, activated *P2Y2* receptors transactivate VEGF receptor 2 (VEGFR-2), suggesting a direct link between extracellular nucleotides and angiogenesis [28]. In addition, activation of *P2Y1* in endothelial cells promotes VEGFR-2 intracellular signaling to stimulate endothelial cell tubulogenesis [3]. *P2Y1* and *P2Y12*- mediated activation of platelets by ADP also increased VEGF concentrations [4]. In turn, loss of *P2Y4* receptor was associated with angiogenic defects and a microcardia phenotype [29]. Adenosine tetraphosphate (Up4A), endogenously released from endothelial cells, was identified to activate migration in smooth muscle cells from the thoracic aorta through activation of *P2Y2* [30] and upregulate *VEGFA* gene and VEGF-A protein levels through purinergic *P2Y* receptors [31]. These findings clearly show that extracellular nucleotides are involved in angiogenesis by targeting purinergic receptors.

The limitation of natural nucleotides is their instability when administered in vivo [13]. However, we have shown that their phosphorothioate analogs are more resistant to enzymatic degradation [14]. Therefore, in the present study, we have investigated the potential proangiogenic action of unmodified and phosphorothioate analogs of nucleotides. 18 compounds were included, 8 of which were unmodified (TMP, UMP, UDP, UTP, CMP, AMP, ADP, ATP), and 10 were phosphorothioate analogs belonging to nucleoside 5′-O-monophosphorothioates (TMPS, UMPS, CMPS, AMPS), 5′-diphosphorothioate (ADPβS, UDPβS), and 5′-triphosphorothioatea (ATPαS, ATPγS, UTPαS, UTPγS).

Our previous studies indicated that nucleoside 5′-O-phosphorothioate analogs, namely UTPγS, UTPαS, and ATPγS, accelerated VEGF release and migration of human HaCaT keratinocytes. The potencies were correlated with the highest *P2Y2* receptor expression by the HaCaT cell line [15]. In the present study, we tested the ability of a wide range of unmodified nucleotides (AMP, ADP, ATP, UMP, UDP, UTP, CMP, TMP) and their phosphorothioate analogs (AMPS, ADPβS, ATPαS, ATPγS, UMPS, UDPβS, UTPαS, UTPγS, CMPS, TMPS) to stimulate secretion of VEGF from fibroblasts. Additionally, the potency of nucleoside 5′-O-monophosphorothioates (TMPS, UMPS, CMPS, AMPS) was assessed in keratinocytes because previously we did not evaluate their activity in HaCaT cells. Our research was primarily focused on TMPS, UMPS, CMPS, and AMPS since earlier we have shown that nucleoside 5′-O-monophosphorothioates can also act as *P2Y* ligands [15,18]. We have evidenced that UMPS, CMPS, and AMPS act as *P2Y14* agonists while TMPS serves as its antagonist [18]. Moreover, we have identified TMPS and UMPS as *P2Y6* agonists [15]. On the other hand, AMPS appeared inactive as a *P2Y6* agonist. However, it acted as a partial agonist of the *P2Y11* receptor [32].

Previously, we also confirmed that the *P2Y2* gene was the most abundant nucleotide receptor subtype in HaCaT cells. *P2Y6* and *P2Y1* expression were shown to be about two times lower. We also found mRNA for *P2Y4* and *P2Y11*, as well as *P2Y12*, *P2Y13,* and *P2Y14* [15]. We identified a completely different expression profile of *P2Y* in HUVECs and HDF cells. Among *P2Y* expressed in HUVEC cells, the highest expression level was noted for the *P2Y6* gene. The analysis also showed relatively high *P2Y1* and *P2Y14* transcripts and a lower *P2Y4*, *P2Y2*, and *P2Y11*. The research also confirmed the expression of *P2Y1*, *P2Y2*, *P2Y4*, and *P2Y6*, while in the case of *P2Y1* and *P2Y6,* the level of detected transcripts was the highest. The high level of *P2Y6* expression in human endothelial cells and fibroblasts was an additional reason to involve nucleoside 5′-O-monophosphorothioates in this study.

We initially evaluated the influence of tested nucleotides on the growth of human keratinocytes, fibroblasts, and endothelial cells. The unmodified nucleotides and their phosphorothioate analogs showed different abilities to stimulate cell survival and/or proliferation depending on the cell model used. The obtained results suggest that the tested compounds exert the most significant influence on the survival of the HaCaT line. UTP and UTPαS were the most effective in intensifying the metabolic activity of HaCaT cells and, to a lesser extent, in stimulating their proliferation. UDP and UDPβS, agonists of the *P2Y6* receptor, have also shown stimulating activity among other uridine nucleotides. We have also proven the expression of this receptor in HaCaT cells, so it can be hypothesized that *P2Y6* participates in the stimulation of cell survival. This hypothesis can also be supported by the fact that UTP and UTPαS, which were most active in these cells, could be hydrolyzed to UDP and UDPαS, respectively, and then activate the *P2Y6* receptor. UTPγS, the most resistant to hydrolysis among uridine triphosphates tested, showed much lower activity. Particularly noteworthy is the activity of TMPS, UMPS, CMPS, and AMPS, which stimulated both the survival and proliferation of HaCaT keratinocytes. Unmodified nucleoside monophosphates did not exhibit such effects. Contrary to the HaCaT, the impact of the tested nucleotides on the survival and proliferation of HDFs fibroblasts was less profound. The highest stimulation level observed for these cells was not higher than 20%. Some compounds, particularly ADP and ATP and their phosphorothioate analogs, showed a significant effect in HUVECs, with ATPγS being the most effective, probably activating *P2Y2* and/or *P2Y11* receptors. Literature data indicate that *P2Y11* is a more likely candidate because the phosphorothioate analog of ATP is a more potent receptor agonist than unmodified ATP [33]. A slight stimulation was also observed in HUVECs cultured in the presence of nucleoside 5′-O-monophosphorothioates.

Our previous studies revealed the high activity of UTPγS, UTPαS, and ATPγS to stimulate the secretion of the vascular endothelial growth factor by the HaCaT keratinocytes [15]. In this study, we demonstrate that stimulation of VEGF-A secretion is also observed in keratinocytes in the presence of nucleoside 5′-O-monophosphorothioates, mainly AMPS and TMPS. Surprisingly, nucleoside 5′-O-monophosphorothioates were the most potent concerning the production of VEGF-A by fibroblasts. These observations again support the hypothesis that nucleoside 5′-O-monothiophosphates may act as ligands for *P2Y* nucleotide receptors. The unmodified nucleoside monophosphates did not induce similar effects.

TMPS appeared to be the most promisin concerning VEGF-A secretion among the nucleotides studied. We showed previously that in a stable *P2Y14*-expressing HEK293T cell line, TMPS inhibited UDPG-induced activation of the *P2Y14* receptor [14]. TMPS also suppressed UDPG-evoked degranulation in antigen-sensitized RBL-2H3 mast cells. Importantly, UMPS, CMPS, and AMPS acted as activating compounds for the *P2Y14* receptor. On the other hand, our studies also proved that TMPS stimulated cellular migration targeting the *P2Y6* receptor [20]. Here, we show the preliminary results that confirm the proangiogenic activity of TMPS under in vivo conditions. One can hypothesize that this activity is correlated with their binding to and activation of the *P2Y6* receptor. However, activation of receptors by 5′-O-monophosphorothioates may not be the only signaling pathway involved in the formation of new blood vessels during our experiments with Wistar rats. Another purinergic signaling cascade mediates the hydrolysis of ATP into ADP, AMP, and finally, adenosine which activates P1 receptors responsible for promoting many essential processes, including angiogenesis and VEGF release. A vital component of this purinergic system is the enzyme ecto-5-nucleotidase (CD73), which catalyzes the last step in the extracellular metabolism of ATP to form adenosine. A recent review by Alcedo et al. presents this enzyme as a critical regulator of cellular homeostasis, stress responses, injury, and disease mechanisms across many tissues [34]. While AMP is the primary substrate of the enzyme, CD73 has been reported to hydrolyze also other 5′-nucleoside monophosphates [35]. Although the activity of CD73 towards 5′-nucleoside monophosphorothioates has not been reported so far, one can suggest that these compounds may be relatively inefficient in enzyme inhibition. Therefore, this alternative signaling pathway CD73—adenosine—P1 receptors should not be influenced by 5′-NMPS and could participate in forming new blood vessels at the BNC scaffold surface.

The ability of TMPS and other nucleotides to augment VEGF secretion is of particular importance. However, recent research suggests that VEGF may potentially act as a profibrotic mediator. During the final phases of wound healing, fibroblasts deposit and restructure collagen to rebuild the dermal layer of the skin, eventually generating a mature scar. In most circumstances, a normal scar will form; however, when fibroblasts create an excessive amount of collagen, aberrant scars such as hypertrophic scars and keloids might occur [36]. Several studies have found a relationship between scar formation and robust angiogenesis. Although the particular mechanisms through which VEGF promotes scar formation are unknown, endothelial cell stimulation or inflammation amplification, as well as possible effects on fibroblasts, have been proposed [37]. Some reports demonstrate the significant contribution of extracellular nucleosides and nucleotides in promoting abnormal matrix production and generate questions on the risk of hypertrophic scars. Excessive ligand-mediated activation of specific nucleoside and nucleotide receptors gives the tissue microenvironment a surplus of signals, promoting abnormal replication of smooth muscle cells, fibroblasts, and myofibroblasts pathologic matrix deposition causing fibrosis [38]. Because we observed that the proliferation of human dermal fibroblasts was only slightly stimulated by most of the nucleotides tested, one could suggest their role in decreasing the risk of hypertrophic scars. However, at present, this field is at an initial stage and many developments have still to be made.

Another issue that has to be emphasized in hypertrophic scar formation is connected with mast cells activation, which leads to the release of newly synthesized mediators [39]. Histamine released from mast cells increases capillary permeability and vasodilation, allowing neutrophils to enter the wound and other mediators such as IL-4 and basic fibroblast growth factor to increase fibroblast proliferation [40]. In addition, mast cell tryptase stored in granules of activated mast cells has chemotactic and mitogenic effects on fibroblasts and participates in collagen synthesis, contraction, and differentiation into myofibroblasts [41]. Because our previous studies demonstrated that TMPS has a unique ability to antagonize *P2Y14* receptor in mast and thereby inhibited mast cell degranulation [18], one can presume that this activity may contribute to anti-profibrotic activity.

The possible application of nucleoside 5′-O-phosphorothioate analogs in the treatment of chronic wounds is also questionable. In general, we see no need for nucleotides to be applied to acute injuries since an initial powerful angiogenic response is characteristic for normally healing acute wounds [42]. Recently, topically applied adenosine-5′-diphosphate was shown to improve the wound healing of diabetic mice but not of healthy nondiabetic animals, which are already highly competent in tissue repair [43]. However, inadequate angiogenesis has been linked to the pathology of chronic wounds, including diabetic wounds, pressure ulcers, and wounds of individuals with venous stasis disease [42]. Several studies have also demonstrated substantial differences with angiogenesis between younger and older people during wound healing [44]. Therefore, the application of TMPS in chronic wounds should be considered especially that we demonstrated that nucleoside 5′-O-phosphorothioate analogues are better candidates to overcome hyperglycaemia-induced impairment of angiogenesis as compared to unmodified counterparts [15]. Besides, an increase in ecto-nucleotidase activity was observed in patients with diabetes and associated pathologies [44]. These observations strongly support the advantage of nucleoside 5′-O-phosphorothioate analogs over unmodified nucleotides also in terms of their increased stability and resistance toward the ecto-5′-nucleotidase present at the cell surface [45].

In this way, we provide new indications that nucleoside 5′-O-phosphorothioate analogs can cause an improvement in the healing of chronic wounds, especially those related to advanced age and diabetes. However, relevant scientific and clinical questions on the role of phosphorothioate nucleotides in wound healing are awaiting an answer. We are aware that the findings of this study have to be seen in the light of some limitations. Several studies suggest that some capillaries formed in early healing wounds are not highly functional, e.g., effectively perfused. Besides, new capillaries produced under high proangiogenic pressure situations, such as in solid malignancies, were shown to be leaky and often ineffective in delivering adequate blood flow [42]. Therefore, further studies assessing if phosphorothioate nucleotides such as TMPS stimulate the formation of functional vasculature with proper perfusion are necessary. More work has to be done with in vivo studies and clinical testing. Finally, characteristics of the patient, such as age and other conditions such as diabetes that might increase the risk of wound healing complications, would have to be also carefully considered.

In summary, the results of this research demonstrate that the presence of extracellular nucleotides, especially phosphorothioate analogs, induces the proangiogenic behavior of fibroblasts, keratinocytes, and endothelial cells. Among investigated compounds, the activity of thymidine 5′-O-monophosphorothioate seems very promising (Figure 6). Our study provides evidence for TMPS-mediated efficient release of VEGF-A from human dermal fibroblasts and keratinocytes accompanied by the augmented expression of *MMP2* and *MACF1* in HUVECS. TMPS also stimulated both the survival and proliferation of HaCaT cells and HDFs and the formation of new blood vessels in a rat model with subcutaneously implanted TMPS-soaked BNC biomaterial. We propose that thymidine 5′-O-monophosphorothioate may be a new candidate for accelerating healing in cutaneous wounds, especially since we have proven previously that TMPS is the antagonist of *P2Y14* involved in inflammatory signaling. These findings suggest that TMPS possesses a promising therapeutic potential in vascular diseases, particularly for patients with chronic skin wounds. However, relevant scientific and clinical questions on the role of phosphorothioate analogs of nucleotides in wound healing are awaiting an answer.

## Figures and Tables

**Figure 1 ijms-23-00238-f001:**
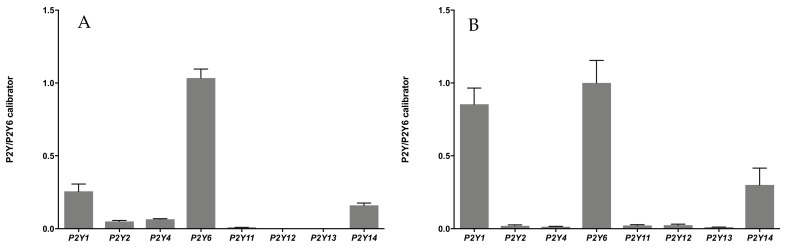
Comparison of expression patterns of genes encoding *P2Y* receptors in HUVECs (**A**) and HDF (**B**) cells. The expression analyzed by the ΔΔC_t_ method was normalized to *GAPDH* and samples were calibrated by *P2Y6*. Bar graphs show the results from at least three independent experiments for each receptor (the mean ± SD).

**Figure 2 ijms-23-00238-f002:**
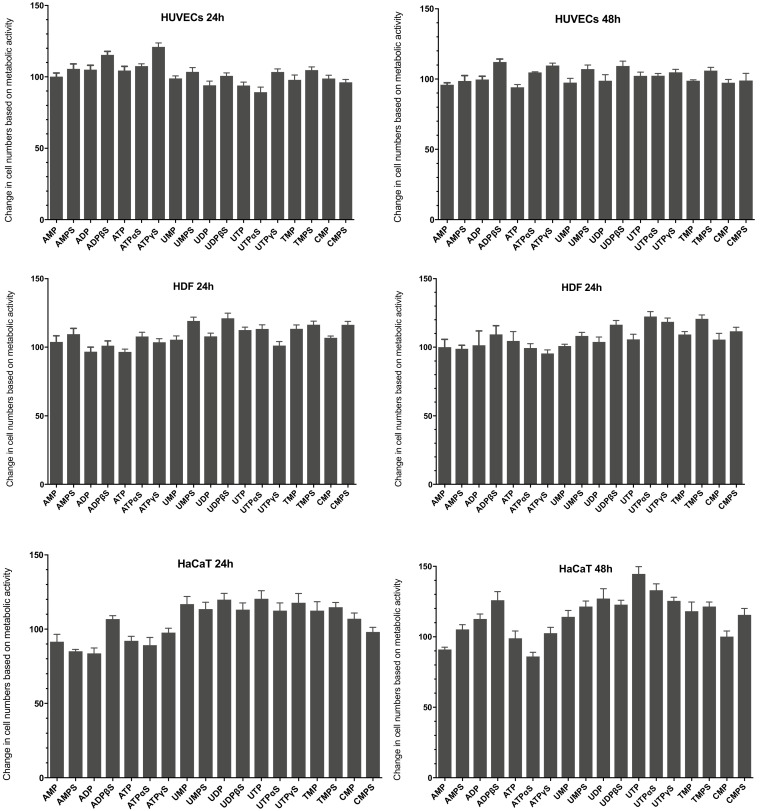
The effect of nucleotides and their phosphorothioate analogs on the metabolic activity of HUVECs, HDFs, and HaCaT cells after 24 and 48 h of incubation. Cell viability was analyzed using the PrestoBlue^®^ test (Life Technologies, Carlsbad, CA, USA) for a concentration of 100 µM of the tested compounds. Results are presented as mean values derived from at least three independent experiments ± SD.

**Figure 3 ijms-23-00238-f003:**
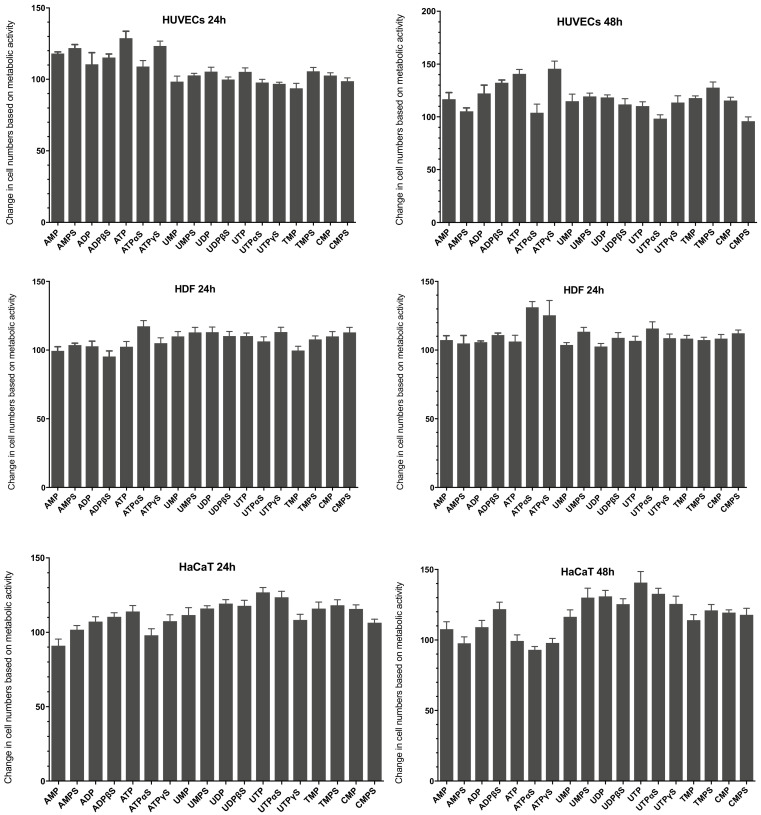
The effect of nucleotides and their phosphorothioate analogs on the proliferation of HUVECs, HDFs, and HaCaT cells after 24 and 48 h of incubation. Cell viability was analyzed using the CyQUANT^®^ (Life Technologies, Carlsbad, CA, USA) test for a concentration of 100 µM of the tested compounds. Results are presented as the mean value derived from at least three independent experiments ± SD data.

**Figure 4 ijms-23-00238-f004:**
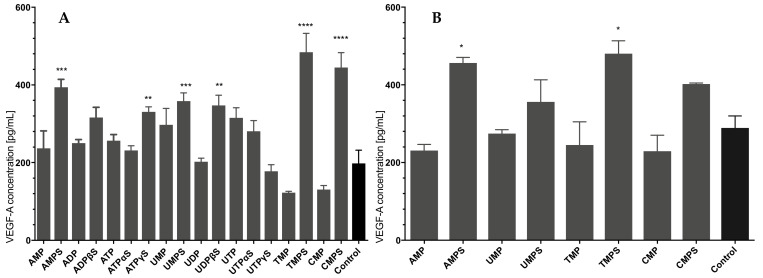
The effect of nucleotides and their phosphorothioate analogs on the VEGF-A secretion in HDF (**A**) and HaCaT (**B**). VEGF concentration was assessed using ELISA for 100 μM concentrations of tested compounds after 24 h of incubation. Data represent the means ± SD from at least three independent experiments; * *p* < 0.05, ** *p* < 0.01, *** *p* < 0.001, **** *p* < 0.0001, vs. unstimulated control cells.

**Figure 5 ijms-23-00238-f005:**
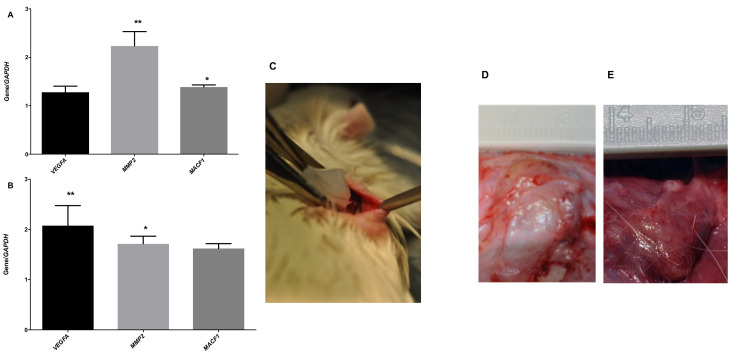
The influence of TMPS on *VEGFA*, *MMP2*, *MACF1* expression in HUVECs (**A**) and HDF cells (**B**) and on angiogenesis in vivo. The subcutaneous implantations of BNC scaffolds were performed on the dorsal region of Wistar rats (**C**). Control BNC (**D**) and BNC with TMPS (**E**) were resected after 30 days, and macroscopic pictures were taken. The gene expression was analyzed by the ΔΔC_t_ method and was normalized to *GAPDH* for 100-μM concentrations of TMPS. Data represent the means ± SD from at least three independent experiments; * *p* < 0.05, ** *p* < 0.01 vs. unstimulated control cells.

**Figure 6 ijms-23-00238-f006:**
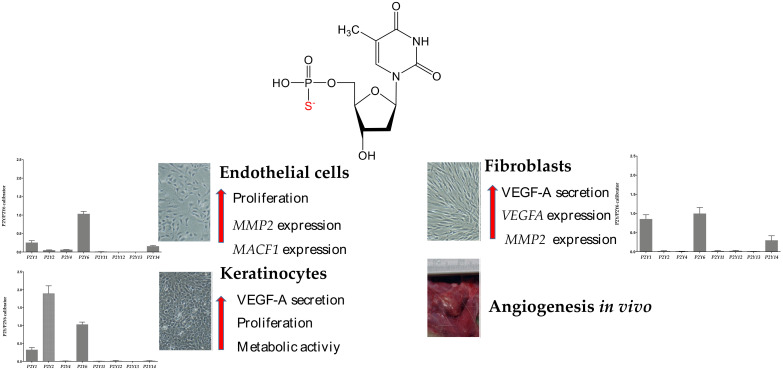
TMPS affects the proangiogenic behavior of fibroblasts, keratinocytes, and endothelial cells through stimulation of (1) VEGF-A secretion from HaCaT keratinocytes and human dermal fibroblasts, (2) *VEGFA* expression in fibroblasts, (3) *MMP2* expression in HUVECs and HDFs, (4) *MACF1* expression in HUVECs, (5) proliferation of HUVECs and HDFs, and (6) metabolic activity in HaCaT. TMPS also promotes host-mediated angiogenic response in vivo. The expression pattern of *P2Y* receptors in cells under study is also shown.

**Table 1 ijms-23-00238-t001:** Sequence of primers used in RT-qPCR reaction.

Gene	NCBI Reference Sequence	Forward Primer	Reverse Primer
*P2Y1*	NM 002563	CGGAAAGTTATCCGCGGCGGT	GGGCTATCGGGCAAGCCAGC
*P2Y2*	NM 176072	GAGGAGCCCCTTGTGGCAGC	CACGCCCAGCCTCCAGCATTTT
*P2Y4*	NM 002565	TGCTGGGCTTGGGCCTTAACG	GGCCGTTGCATCCCAGGGTC
*P2Y6*	NM 176798	ATGCCTGCTCCCTGCCCCTG	GGCGAAGTCGCCAAAGGGCC
*P2Y11*	NM 002566	ACAGAGCGTATAGCCTGGTG	TGTGGTAGGGCACATAGGA
*P2Y12*	NM 022788	TCTGCGCCTGGTAACACCAGTCT	AACAGGACAGTGTAGAGCAGTGGGA
*P2Y13*	NM 176894	GGTCAGCAAGACCTCTGAAA’	AAGGCATTGCTGAGTAGGTG
*P2Y14*	NM 001081455	TGAATCCTGCTCTCAGAACC	AGGCTCATCACAAAGTCAGC
*GAPDH*	NM 002046	AAGGCTGGGGCTCATTTGCAGG	GCCAGGGGTGCTAAGCAGTTGG

## Data Availability

Data from this study is secured in a password protected computer in the Institute of Molecular and Industrial Biotechnology, Lodz University of Technology, Poland.

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
