# Peer review of "Extracellular Nucleotides Affect the Proangiogenic Behavior of Fibroblasts, Keratinocytes, and Endothelial Cells"

_ijms, 2021, doi:10.3390/ijms23010238_

Round 1

Reviewer 1 Report

I think that the topic is of interest, but I have several questions:

-What is the real potential clinical benefit? Please discuss

-Do you exspect differences between acute and chronic wounds? Please discuss

-Do you exsprect differences between young and old? Please discuss

-What about the risk of developing hypertrophic scars? Please discuss

-The limitations of this study are not really mentioned and discussed in a profound way. Please do so.

Reviewer 2 Report

In principle, the manuscript is potentially interesting as it deals with the treatment of chronic wounds a current and important topic. The authors study the effects of a series of unmodified and phosphorothioate nucleotide analogs on the secretion of VEGF from keratinocytes and fibroblasts, as well as their influence on the viability and proliferation of keratinocytes, fibroblasts, and endothelial cells. In addition, the expression profiles of genes encoding nucleotide receptors in 20 tested cell models were also investigated. Finally they defined thymidine 5′-O-21 thiomonophosphate (TMPS) as a positive regulator of angiogenesis and confirmed the proangiogenic potency of TMPS in vivo. 

In my opinion, the manuscript could be more interesting if it included a pertinent iconography relating to the cell types involved as well as those relating to evoked angiogenesis. In this last case the authors show macroscopic but not microscopic photographs. 

Round 2

Reviewer 1 Report

No further comments

Reviewer 2 Report

The authors responded promptly to my questions. The manuscript is now ready for a possible publication